# NMR Investigations of Crystalline and Glassy Solid Electrolytes for Lithium Batteries: A Brief Review

**DOI:** 10.3390/ijms21093402

**Published:** 2020-05-11

**Authors:** Daniel J Morales, Steven Greenbaum

**Affiliations:** 1Department of Physics and Astronomy, Hunter College of the City University of New York, New York, NY 10065, USA; dmorales@gradcenter.cuny.edu; 2Ph.D. Program in Physics, CUNY Graduate Center, New York, NY 10036, USA

**Keywords:** NMR, inorganic electrolytes, glassy electrolytes, ceramic electrolytes

## Abstract

The widespread use of energy storage for commercial products and services have led to great advancements in the field of lithium-based battery research. In particular, solid state lithium batteries show great promise for future commercial use, as solid electrolytes safely allow for the use of lithium-metal anodes, which can significantly increase the total energy density. Of the solid electrolytes, inorganic glass-ceramics and Li-based garnet electrolytes have received much attention in the past few years due to the high ionic conductivity achieved compared to polymer-based electrolytes. This review covers recent work on novel glassy and crystalline electrolyte materials, with a particular focus on the use of solid-state nuclear magnetic resonance spectroscopy for structural characterization and transport measurements.

## 1. Introduction

As lithium ion batteries continue to permeate the commercial market, the search continues to produce an all solid-state equivalent with the same or superior performance. While liquid organic electrolytes continue to exhibit high performance and long cyclability, the risk of thermal runaway and inability to utilize Li metal anodes without the risk of dendrite formation are ongoing issues. The use of solid electrolytes would allow for the use of lithium metal anodes, which would nearly double the current energy density seen in Li-ion batteries. This attractive prospect has fueled considerable research into a variety of solid electrolytes—in particular, inorganic solid electrolytes. Inorganics comprise many different families of materials, including glasses, glass-ceramics, garnets, and others, with a wide range of structural and electrochemical properties, as sampled in the table below. Depending on the material, one can achieve ionic conductivities on the order of 10^−7^ to 10^−2^ S/cm at room temperature, as seen in Figure 1, the high end of which is comparable to liquid electrolytes [1]

It is often useful, and indeed important, to study the Li-ion dynamics in these systems and relate it back to the structure, as it may lead to insights into the nature of the ionic conductivity and offer solutions in terms of improving the electrolyte performance.

## 2. Techniques in Nuclear Magnetic Resonance

Of the many analytical techniques available to probe these materials, NMR Spectroscopy is uniquely capable of studying electrolyte structure and ion dynamics, with a variety of pulse sequences and experiments to determine both long- and short-range dynamics. Nuclear spins precess about a static magnetic field $B_{0}$ at a characteristic Larmor frequency ω [3],
$$\omega =-\gamma B_{0}$$
where γ is the gyromagnetic ratio of the nucleus. These spins form a net magnetization in the direction of $B_{0}$ and can be manipulated by application of a smaller resonant rf pulse. The net magnetization relaxes back to equilibrium according to longitudinal and transverse relaxation times T_1_ and T_2_. Inversion and saturation recovery experiments can determine these characteristic relaxation times, which are mediated by local nuclear, and in some cases quadrupolar spin interactions. These relaxation times are functions of both temperature and frequency and have historically been well described by the Bloembergen, Purcell, and Pound (BPP) relaxation theory, which relates relaxation times to rotational correlation times $\tau_{c}$ [4],
(1)$$\frac{1}{T_{1}}=\frac{3\gamma^{4}\hbar^{2}}{10r_{0}^{6}}\left[J\left(\omega \right)+4J\left(2\omega \right)\right]$$
where J(ω), the spectral density function, is the probability of finding a molecule rotating at a given frequency ω,
$$J\left(\omega \right)=\frac{\tau_{c}}{1+\omega^{2}\tau_{c}^{2}}$$

Correlation times exhibit Arrhenius behavior and can be expressed as
(2)$$\tau_{c}\left(T\right)=\tau_{0}e^{\frac{E_{a}}{kT}}$$
allowing one to extract activation energies $E_{a}$ from variable temperature (VT) relaxation measurements. One simple example of a relaxation experiment is the inversion-recovery pulse sequence, where a π pulse rotates the equilibrium magnetization into the negative z-axis and relaxes for a time *t* before a $\frac{\pi }{2}$ pulse is applied and the resulting signal intensity is measured. The intensity is dependent on both *t* and $T_{1}$ [3],
(3)$$S\left(t\right)=S_{0}\left(1-2e^{-\frac{t}{T_{1}}}\right)$$

One caveat of this experiment is the fact that the relaxation can only be found at one frequency, which is dependent on the field strength of the superconducting magnet used. In field cycling (FC) NMR, the magnetic field is varied over a wide range (10 kHz–1 MHz), allowing for measurements of T_1_ as a function of larmor frequency [5,6].

Pulsed Field Gradient (PFG) NMR is another technique which is well-suited for studying ion transport in a system. As the Larmor frequency of individual spins depends on the local magnetic field, the application of a magnetic field gradient can encode the spatial position of the nuclear spins. By encoding the spins position, allowing time for diffusion, and decoding the spins through a reverse gradient, it is possible to determine the self-diffusion coefficient *D* of the probe nucleus [7] through the Stejskal-Tanner equation:(4)$$S\left(g\right)=S_{0}e^{-D{\left(\gamma \delta g\right)}^{2}\left(\Delta -\frac{\delta }{3}\right)}$$
where g is the gradient strength, δ is the gradient pulse length, γ is the nucleus-dependent gyromagnetic ratio, and Δ is the diffusion time. Self-diffusion as a quantity is tied to various transport properties useful in the field of energy storage, such as cation transference numbers and ionic conductivities of materials through the Nernst–Einstein Equation:(5)$$\sigma_{NMR}=\frac{F^{2}}{RT}\underset{i}{\sum }c_{i}D_{i}$$
where *F* is Faraday’s constant, *R* is the ideal Gas constant, *T* is the temperature in Kelvin, and $c_{i}D_{i}$ is the product of molar concentration and self-diffusion coefficient of each charge carrier in a material.

Aside from ion dynamics, NMR is well known for its ability to perform structural characterization, usually through the use of magic angle spinning (MAS) NMR. The NMR Hamiltonian contains many terms (homo and heteronuclear dipolar interaction, first order quadrupolar interaction, chemical shift anisotropy, etc.) that contain a second order Legendre polynomial term $3\cos^{2}\theta -1$. By rotating a sample at a specific angle to the magnetic field, such that this term cancels out (θ≈54.7°), one can average out most anisotropic interactions, leaving only isotropic shifts, and significantly increasing spectral resolution [8]. This technique can also be paired with cross-polarization (CP) NMR [9], where magnetization is transferred from nearby abundant nuclei—usually protons—to less sensitive or abundant nuclei.

This review will focus on recent publications on glassy and crystalline electrolytes studied through the lens of NMR spectroscopy. Although these selected works do not comprise the entirety of NMR research on solid electrolytes, they do represent most of the major techniques and approaches currently used in the field.

## 3. Glassy Electrolytes

Glass-ceramic solid electrolytes have garnered much attention over the past few decades, due to their notable advantages over their crystalline counterparts. They are relatively easier to make into thin films, and their amorphous nature from the lack of a grain boundary allows for more uniform ion conduction and higher ionic conductivity, making it an attractive electrolyte candidate. Most glasses take the form of a mixture of Li_2_S, GeS_2_, SiS_2_, and P_2_S_5_ in various compositions. Depending on the composition and concentration of lithium, one can achieve ionic conductivities ranging from 10^−7^ to 10^−2^ S/cm [2,10].

Many glassy electrolytes are usually prepared by ball-milling rather than through a melt, which leads to difficulties in scaling up. Glasses that come from a melt tend to lack crystalline conduction pathways or interfaces, which are desirable in discouraging Li dendrite formation. Kaup et al. recently reported on the synthesis of a quaternary glass system containing aLi_2_S-γB_2_S_3_-xSiO_2_-zLiI (LIBOSS), which exhibited high room temperature conductivity (~1 mS/cm) [11]. In addition, LIBOSS glasses were prepared solely through a melt, which allows for a high yield and relative scalability compared to ball-milling. Using MAS NMR, the authors studied the relationship between the silica content x with the changes in ionic conductivity.

Figure 2 shows both the activation energy and ionic conductivity of the glass as a function of SiO_2_ content. Initially, the x = 0 glass revealed a significant amount of crystalline LiI (~28 wt%), which may account for the lower conductivity. However, small amounts of silica (0 ≤x≤ 0.25) drastically improve the solvation of LiI, and between 0.25 ≤x≤ 0.75, all of the LiI is dissolved into the glass matrix. Past x = 0.75, however, the solubility limit of LiI and SiO_2_ is reached and it begins to recrystallize. Additionally, the ionic conductivity of LIBOSS peaks at a silica content of 0.25 (2 mS/cm), but decreases and remains steady at higher x, until dropping at x = 0.75, when the LiI crystallizes.

^11^B and ^29^Si MAS NMR, as seen in Figure 3, were utilized to elucidate this trend. The ^11^B spectra of the x = 0 sample revealed the presence of multiple boron-containing tetrahedral units, such as BO_3_, BO_2_S, BS_3_, and BS_4_ [12,13]. In particular, the peaks are −2.9 ppm and −0.5 ppm correspond to BS_4_ with 2–4 bridging sulfur atoms and BS_4_ with 0–1 bridging sulfurs, respectively. As the silica content increases, the intensities of these two peaks swap, confirming the increased presence of isolated boron tetrahedra, and suggesting that BS_4_ plays less of a bridging role compared to SiO_2_. The increase in non-bridging sulfur units forms smaller inorganic units, which increases the solubility of LiI in the glass, and increases the Li^+^ ion density. In addition, a preference for BO_3_ arises with increasing Si content, as boron has a stronger affinity for oxygen over sulfur, leading the sulfur to bond with silicon. This is seen in the ^29^Si spectra, with the presence of two peaks at 7 ppm and −4.5 ppm, corresponding to SiS_4_ and SiOS_3_ units, respectively.

The lithium environment can also give insight into the nature of the Li^+^ ion mobility. At x = 0, the ^7^Li spectra shows a sharp peak at −4.5 ppm, corresponding to LiI, with a broad peak near 0 ppm corresponds to a distribution of Li in proximity to nonbridging sulfur (in x = 0) or a mixture of sulfur/oxygen (x = 0.25, 0.5). The solubility of LiI greatly increases with increasing Si content, leading to a significantly reduced, albeit nonzero signal intensity in the corresponding peak. The inability of XRD to detect this small amount of LiI suggests the existence of nanoaggregates, which would still hinder the ionic conductivity.

As the SiO_2_ content increases, the 0 ppm peak exhibits great motional narrowing, suggesting increased Li ion mobility. In addition, the peak shifts to lower frequency at higher Si content, suggesting a more ionic interaction between Li and the glass.

In general, the addition of SiO_2_ has the effect of increasing the fraction of non-bridging sulfur/oxygen containing units, which results in small inorganic units, allowing for a greater degree of freedom in the glass network and more terminal chalcogenide ions, leading to greater Li^+^ ion mobility. However, at higher SiO_2_ content, the ionic character of these inorganic groups dominates, and lead to a greater electrostatic interaction with Li^+^ ions near the nonbridging chalcogen, which in turn reduces the ionic conductivity.

While MAS is extremely useful at determining the local structure of glassy materials, alternate methods such as relaxometry can probe the Li ion dynamics at various timescales [14]. Marple et al. studied the Li-ion dynamics of Chalcogenide Glasses, composed of Li_2_S, Ga_2_Se_3_, and GeSe_2_ (LGGS) in mixed compositions, ranging from 7.5% to 50% Li [15,16]. They performed electrochemical impedance spectroscopy (EIS) to find ionic conductivities as a function of both composition and temperature, and extracted Li ion activation energies from the subsequent Arrhenius plots. In particular, the highest room temperature DC conductivity was found in the 50–10–40 Li_2_S–Ga_2_Se_3_–GeSe_2_ glass, approximately 10^−4^ S/cm.

^7^Li NMR lineshape analysis was performed on three phases of the LGGS glasses, 40–60 Li_2_S–GeSe_2_, 15–25–60 Li_2_S–Ga_2_Se_3_–GeSe_2_, and 50–10–40 Li_2_S–Ga_2_Se_3_–GeSe_2_, in order to highlight the differing Li ion dynamics with composition and temperature. Static NMR spectra were obtained using a solid echo pulse sequence of ${\left(\frac{\pi }{2}\right)}_{x}-{\left(\frac{\pi }{2}\right)}_{y}$ pulses, in order to clearly study the quadrupolar effects that are commonly seen in ^7^Li.

From the Variable Temperature (VT) Spectra in Figure 4, a broad Gaussian central transition peak and broader satellite peaks are present; the broad central transition is due to homonuclear dipolar interactions between primarily ^7^Li ions. This is corroborated by the fact that the central transition peak linewidth increases with increasing Li content. As the temperature increases, both central and satellite transitions narrow monotonically, which is indicative of averaged motion of the Li ions overcoming the strong dipolar interaction and suggests a random distribution of Li ions in the glass.

Plotting the ^7^Li NMR lineshapes for each glass reveals a characteristic sigmoid shape. The inflection point of these lineshape functions can give insight into how fast Li ions jump in the material, through calculation of the correlation time, $\tau_{c}$. At the inflection point, the Li-ion jump rate can be estimated as $1/2\pi \nu_{inflection}$, where $\nu_{inflection}$ is the inflection point linewidth [17]. Additionally, one can also approximate the activation energy for the Li ion jumps by finding the temperature at which this inflection occurs [17]. Marple et al. calculated hopping frequencies and their corresponding activation energies using this method, which are highlighted in Figure 5. These results were in close agreement with EIS derived hopping frequencies and activation energies. From this agreement, the authors argued that the hopping frequency corresponds to Li ion jumps between an average Li–Li separation distance and were able to estimate the DC conductivity using the Nernst–Einstein relation.

The calculated conductivities were well in agreement with EIS calculated conductivities, which further suggest that the model of intersite Li ion jumps is correct.

It is known that combinations of glass formers can lead to anomalously ionic conductivities, hereafter referred to as the “mixed glass former effect” (MGFE) [12,14,18,19,20]. This effect can either drastically increase or decrease ionic conductivity, known as the positive and negative MGFE, depending on the compositions and concentrations of the glasses mixed. NMR lineshape and relaxation studies have been used to study this effect. Storek et al. looked at a series of glasses containing mixed concentrations sodium borosilicate and sodium borophosphate glasses, known to produce a negative and positive MGFE, respectively [21]. Using ^23^Na NMR spectra, they attempted to correlate short-range structural information with microscopic origins of the MGFE.

From the ^23^Na lineshapes, there was clear motional narrowing observed for all glasses with increased temperature, with clear differences in the inflection point linewidth depending on the composition of the glass. This effect was even more pronounced when looking at spin-lattice relaxation measurements, as shown in Figure 6. The borosilicate glasses 0.33Na_2_O + 0.67[xB_2_O_3_ + (1 − x)2SiO_2_] experienced slower relaxation rates than the borophosphate glasses and decreased with increasing B_2_O_3_ content, experiencing a relaxation minimum around x = 0.6; this suggests a negative MGFE. Conversely, the borophosphate glass 0.35Na_2_O + 0.65[xB_2_O_3_ + (1 − x)P_2_O_5_] experienced an increase in relaxation rates with increasing Boron content, eventually reaching a maximum when x = 0.4, indicating a positive MGFE.

To explain this behavior, the authors attempted to model the relaxation data using only one tunable parameter, the mean activation energy E_a_. In glasses, it is generally agreed that there is a broad range of energies, so a gaussian distribution of energies was assumed. Using expressions for the relaxation rates and linewidths dependent on this model, the authors were able to accurately and simultaneously fit the data [22,23] and extracted mean activation energies on the order of 0.1–1 eV, depending on the B_2_O_3_ content. Comparing these activation energies with those found from conductivity measurements shows that conductivity-measured energies are consistently lower than their NMR-derived counterparts. The authors proposed that conductivity measurements were unable to detect below a certain threshold activation energy. Proposing an NMR-based model using the gaussian energy distribution, they calculated the threshold activation energy to be in agreement with conductivity measurements, underscoring the utility of the model.

In addition, ^11^B MAS NMR, detailed in Figure 7, was used to determine the fraction of network forming units (NFUs) for Si^(3)^, Si^(4)^, B^(2)^, B^(3)^, and B^(4)^ in the borosilicate glasses, depending on B_2_O_3_ content, and compared to the Yun and Bray model [24,25], which was in good agreement. From the model, it was thought that the Na ions preferred to cluster around Si atoms, and the addition of boron reduces this clustering leading to slower Na ion dynamics and lower conductivity, i.e., a negative MGFE. This effect was also seen in other mixed glass systems, such as sodium germanophosphate glasses (NGP) [26], further lending credibility to the model.

## 4. Glass Ceramic Electrolytes

Aside from pure-glass electrolytes, much work has been done studying glass-ceramics, which simultaneously exhibit both amorphous and crystalline phases [27]. Typical glass-ceramics exhibit comparable ionic conductivities to glasses—up to 10^−2^ S/cm at room temperature [28,29]—which is possibly due to the reduction of grain boundary resistance during the crystallization process.

Gobet et al. studied one such glass-ceramic, Li_3_PS_4_, which is known to exhibit a high ionic conductivity of ~10^−3^ S/cm [30,31] and is chemically stable versus metallic lithium [32]. Li_3_PS_4_ can exhibit different phases and under conventional ball-milling techniques presents as a γ phase, which exhibits a low ionic conductivity [33]. However, a new synthesis technique, discovered by Liang et al., revealed that Li_3_PS_4_ can be formed through the desolvation of THF in a mixture of ${\mathrm{Li}}_{2}S$ and $P_{2}S_{5}$, leaving behind a nanoporous structure. This structure was shown to exhibit an ionic conductivity three orders of magnitude higher than γ-Li_3_PS_4_ and is referred to as β-Li_3_PS_4_. As this novel synthesis approach appeared to change the phase of the glass-ceramic, solid state NMR measurements were performed to understand the structural evolution of β-Li_3_PS_4_.

^1^H, ^31^P, and ^6^Li and ^7^Li MAS NMR experiments were performed to understand and characterize the synthesis process. Samples of Li_2_S–P_2_S_5_ precursors in THF were treated at various temperatures to drive off the THF solvent and observe the formation steps of the final product. Spectra were compared to those of pure γ-Li_3_PS_4_ and β-Li_3_PS_4_ (prepared with THF). Single-pulse measurements were performed at a range of spinning speeds, from 4 kHz to 17 kHz; ^1^H–^31^P and ^1^H–^7^Li cross polarization (CP) experiments were also performed at 4 kHz to study potential interactions between the solvent and probe nuclei. PFG experiments were also performed at 100 °C to determine the ^7^Li diffusion coefficients.

From the ^1^H spectra in Figure 8, the presence of a peak at ~3 ppm confirms the presence of THF in the sample, when compared to liquid spectra of pure THF. At higher treatment temperatures, the intensity of this peak is significantly diminished, suggesting the removal of THF, as expected.

The ^31^P spectra in Figure 8 paints a more interesting picture. Li_3_PS_4_ with THF exhibits three separate peaks, suggesting several chemical environments surrounding the PS_4_^3−^ tetrahedra. As the sample is heat-treated, these peaks give way in favor of a broad amorphous component near 85 ppm and at 100 °C, only the broad component remains. This agrees with the proton spectra at the same temperature, suggesting that there is no longer any THF to interact with the PS_4_^3−^ tetrahedra. At even higher temperatures, a separate peak emerges which suggests the presence of isolated PS_4_^3^ tetrahedra associated with the β-Li_3_PS_4_ phase, which could not be determined by long-range analytical techniques such as XRD. In the ^7^Li spectra, the presence of spinning sidebands suggested a greater quadrupolar interaction due to the presence of electric field gradients that are not effectively averaged by rapid ionic motion. These sidebands greatly decreased in intensity when the samples were heat-treated, which is indicative of increased ion mobility.

Through CP-MAS experiments, the authors were able to probe the extent of the THF’s interaction and proximity to other probe nuclei, as CP is mediated through the nuclear dipole-dipole interaction. In the unheated Li_3_PS_4_•3THF mixture, CP experiments for ^31^P and ^7^Li were nearly identical to their single pulse counterparts, suggesting that both nuclei were surrounded by nearby proton-containing THF molecules. For the sample treated at 70 °C, comparison of ^31^P CP and single pulse experiments, as seen in Figure 9, revealed the presence of a broad peak that does not couple to the THF molecules near 84 ppm. This peak is known to belong to a combination of PS_4_^3−^ and PS_3_O^−^ peaks [34,35], which suggests that THF molecules decomposed during the heating process, leading to an O–S exchange.

^7^Li PFG experiments revealed the self-diffusion coefficient to be on the order of 10^−13^ m^2^/s. Comparison with the EIS and NE derived diffusion reveals a D on the order 10^−12^ m^2^/s, which is a clear discrepancy. This could be due to multiple factors, including structural evolution of the sample during the course of the experiments, or underestimation of the NMR diffusion coefficients due to the long diffusion time needed for the experiment (~500 ms).

Another recent work on glass-ceramics by Murakami et al. looked at Li_7_P_3_S_11_, known to exhibit a room temperature conductivity of 10^−3^ S/cm [36]. Li_7_P_3_S_11_ is a metastable material, formed by annealing (Li_2_S)_70_(P_2_S_5_)_30_ at temperatures over 500 K. From XRD analysis, it is known that Li ions in the material are situated between PS_4_ tetrahedra and P_2_S_7_ ditetrahedra. In addition, they are known to conduct through two pathways—a Li stable and a Li metastable region [37]. As an electrolyte, the Li_7_P_3_S_11_ system has been studied previously from the NMR perspective [38]. In order to further study the origins of the high ionic conductivity, the authors employed solid state ^6/7^Li and ^31^P NMR.

The authors prepared three samples for NMR measurements: (Li_2_S)_70_(P_2_S_5_)_30_ glass-ceramic (70 gc), (Li_2_S)_75_(P_2_S_5_)_25_ glass-ceramic (75 gc), and (Li_2_S)_70_(P_2_S_5_)_30_ glass (70 g). MAS single pulse spectra, as well as T_1_ saturation recovery measurements, were performed as a function of temperature, ranging from 250 K to nearly 400 K.

From the ^7^Li lineshapes, it is apparent that motional narrowing of the spectra takes place with increased temperature, as the Li–Li dipolar couplings and Li quadrupolar interaction are averaged out. This motional narrowing gives rise to the presence of multiple convoluted peaks. Further peak analysis performed on the 70 gc sample, highlighted in Figure 10, revealed the presence of three peaks, two narrow and one broad. The narrow peaks were attributed to mobile Li ions in a crystalline structure, while the smaller broad peak corresponded to more less mobile Li ions in the structure.

^6/7^Li and ^31^P T_1_ measurements, as seen in Figure 11, allowed the authors to calculate the associated activation energies for 70 gc, 18 kJ/mol. In addition, the authors found that the phosphorus nuclei bore the same temperature dependence as lithium nuclei at lower temperatures, but diverged after 300 K, suggesting independent motion. To understand the role of this motion and its contribution to the ionic conductivity, the authors studied the ^31^P MAS spectra of 70 g and 75 gc as a function of temperature.

They noticed that the lineshapes of the ^31^P spectra did not significantly narrow with increased temperature, which seems to suggest that the rotational motion of the $P_{x}S_{y}$ groups is responsible for the anomalous relaxation at higher temperatures, rather than translational motion. In 70 gc, there are three distinct ^31^P sites, corresponding to P_2_S_6_^4−^, P_2_S_7_^4−^, and PS_4_^3−^. The lineshapes for all three peaks remained temperature independent until 300 K, after which motional narrowing was observed for all but the P_2_S_6_ peak. Additionally, ^31^P–^31^P radio-frequency-driven recoupling (RFDR) experiments were performed to examine the line widths further. Off-diagonal peaks in the 2D spectrum merged into the P_2_S_7_^4−^ peak at higher temperatures, and the authors argued that the peak could not be attributed to a particular asymmetric ditetrahedra structure of P_2_S_7_^4−^. This suggested a distribution of local structures associated with P_2_S_7_^4−^ and that the dynamical motions of the ditetrahedra allowed for a pathway for Li ions to conduct at a lower energy barrier, leading to higher conductivity.

The ${\mathrm{Li}}_{10}{\mathrm{GeP}}_{s}S_{12}$ (LGPS) family of lithium superionic conductors (LISICONs) have been extensively studied since it’s 2011 discovery by Kanno et al. [39] due to their high room-temperature conductivities (~10^−2^ S/cm for LGPS) resulting from the formation of a 1D conduction pathway for Li^+^ ions. In general, the family can be classified as Li_11−x_M_2−x_P_1+x_S_12_, where M can be Si, Ge, or Sn. Though sporting high conductivity, these materials exhibit poor stability at the interface against metallic Li due to the formation of interphases containing Li_3_P, Li_2_S, and Li–Ge alloys, which promotes reduction of the electrolyte and can lead to a short-circuit [40]. To counteract this, Harm et al. synthesized a new LGPS-like glass ceramic, Li_7_SiPS_8_ (LSiPS), and characterized its structure and Li-ion mobility through ^29^Si and ^31^P MAS, and ^7^Li PFG NMR [41]. The authors synthesized LSiPS through mixing stoichiometric amounts of Li_2_S, Si, red P, and sublimed S and annealing the mixtures at differing temperatures for five days, yielding both tetragonal and orthorhombic phases.

The authors first attempted to characterize the structure of LSiPS through ^31^P MAS, the results of which are shown in Figure 12. In tetra-LSiPS, they found two peaks at ~73 ppm and 95 ppm, corresponding to phosphorus nuclei inhabiting the 2b and 4d sites, respectively, while one main peak at ~88 ppm was detected for ortho-LSiPS, corresponding to the 4c position. In addition, the authors identified an amorphous side phase in the tetra-LSiPS sample, constituting almost 20% of the signal intensity.

Further evidence of an amorphous phase appears in the ^29^Si spectra of the tetra-LSiPS sample. The main peak near 10 ppm corresponds to an expected substitution of Si into the phosphorus 4d sites, but a small peak (~3% signal intensity) near 0 ppm further lends credit to the presence of the amorphous phase. As this peak could not be detected in XRD measurements, it was concluded that tetra-LSiPS crystals were embedded in a glassy matrix, indicative of a glass-ceramic. By comparing the relative NMR signal intensities to XRD-derived weight percentages, the authors proposed the amorphous phase to be Li_3.2_Si_0.2_P_0.8_S_4_. In addition, this phase only appears at samples prepared at intermediate temperatures, which suggests that the glassy phase plays a role in the transition from ortho-LSiPS to tetra-LSiPS. As this proposed structure does not share the same stoichiometry with the crystalline phase, it is thought that this amorphous phase is the result of a peritectic phase separation of the tetragonal phase and thus may be less stable than tetragonal LGPS.

To understand the nature of Li-ion diffusion in this novel material, the authors employed variable temperature ^7^Li PFG NMR at two different diffusion times (10 ms and 100 ms) and calculated the respective activation energies, the results of which are plotted in Figure 13. Using diffusion coefficients at room temperature, the authors calculated the diffusion length at both time scales through a first-order approximation, $r_{rms}=\sqrt{2D_{NMR}\Delta_{\mathrm{NMR}}}$, and found that the distances were both on the order of 200 nm, which suggests a fast diffusion process over a small range. Assuming that the crystalline phase primarily contributed to conductivity, the authors assumed the diffusion radius to be the mean size of the crystallites.

In addition, the authors deduced that diffusion at shorter timescales was higher and exhibited a lower activation energy, suggesting at an overall mixed intergrain and intragrain diffusion process, in agreement with the assumption of crystallites embedded in an amorphous matrix. Both NMR-calculated and EIS conductivities were on the order of 1 mS/cm, lower than tetragonal LGPS, which suggests that the inclusion of the glassy phase comes at the cost of increased impedance.

## 5. Argyrodite Electrolytes

The cubic mineral argyrodite, with the composition Ag_8_GeS_6_, is known to conduct Ag^+^ ions very easily. Replacing Ag with another cation, in this case Li^+^, will still preserve the cubic structure. As a result, much research has been done into argyrodite-type electrolytes for Li batteries [42,43,44], with Li_7_PS_6_ emerging as a possible candidate [28]. However, it loses its cubic structure at room temperature and above; partial substitution of S for a halogen has been shown to preserve the cubic structure and ionic conductivity [43,45]. Typical Argyrodite electrolytes are of the form Li_6_PS_5_X, where X is typically Br, Cl, or I. Hanghofer et al. studied these halogen-containing argyrodites, in order to understand the nature of Li ion conductivity and its dependence on the choice of halide ion [46]. The authors studied Li_6_PS_5_I, Li_6_PS_5_Br, and Li_6_PS_5_Cl, and performed static ^7^Li relaxation experiments to determine activation energies and jump rates, as well as ^31^P MAS to understand structural characteristics of each material.

Figure 14 displays the ^31^P and ^7^Li MAS spectra of all three materials. From the ^31^P MAS spectra, it was clear that the halide ion plays a large role in the structure of the material. Li_6_PS_5_I exhibited a single narrow peak, suggesting an ordered crystalline environment around the phosphorus nuclei [47,48], while Li_6_PS_5_Cl revealed barely resolvable peaks, suggesting a considerable amount of anion disorder; Li_6_PS_5_Br presented multiple disordered peaks as well.

In addition, ^7^Li relaxation measurements, as seen in Figure 15, show that the degree of cation disorder is also dependent on the choice of halide. Using modified BPP models, the authors calculated the activation energies of each material, both at low temperature and high temperatures. It was clear that Li_6_PS_5_Br exhibited the lowest activation energies on both temperature flanks of the relaxation peak, followed by Li_6_PS_5_Cl, with Li_6_PS_5_I exhibiting the highest energy barriers; this was further confirmed by EIS measurements, pinning Li_6_PS_5_I as having the lowest room temperature conductivity of the series. From this data, one could not ignore the relationship between anion disorder and cation mobility.

To try and explain this phenomenon, the authors employed lineshape analysis of ^7^Li spectra to observe the onset of motional narrowing, which is expected to arise due to increased intercage jumps between Li sites in the lattice. While this lineshape averaging was seen at low temperatures (~250 K) for Li_6_PS_5_Br and Li_6_PS_5_Cl, the Li_6_PS_5_I spectra exhibited considerably less narrowing, achieving the same narrow lineshapes nearly 100 °C higher than other two materials. This suggests that Li_6_PS_5_I exhibits much higher activation energy barriers to account for the slower jump process, which could be due to the rigid anion structure seen via the ^31^P spectrum.

## 6. Garnet Electrolytes

Garnet-type oxide electrolytes have received much attention over the past few years, ever since the discovery of Li_7_La_3_Zr_2_O_12_ (LLZO) by Murugan et al. [28,49]. These garnets are cubic in structure, maintain high working voltages against Li [50], and exhibit high ionic room temperature conductivities, on the order of 10^−4^ S/cm in LLZO. Many recent works in the field of garnet electrolytes have involved doping the Li or Zr sites in LLZO, in order to improve the ionic conductivity [51,52,53,54,55,56], and NMR is well suited at studying the subsequent structural changes and defect chemistries [55,57] as well as Li ion dynamics.

One group, Larraz et al. sought to elucidate the distribution of Li ions in LLZO, as the ^7^Li isotropic chemical shifts tend to fall in a narrow window in these materials, making it difficult to resolve separate peaks [58]. This difficulty is further compounded by the presence of ^7^Li dipolar broadening, which is still present when performing MAS. In addition, the exposure of these crystals to air allows for proton-lithium exchange and for new secondary phases and impurities to form, such as LiOH and Li_2_CO_3_ which further compound the problem [59,60]. To work around this, the group synthesized LLZO using enriched ^6^Li, to allow for much more resolvable spectra. These crystals were then purposefully exposed to air by annealing to allow for proton-Li exchange in order to vary the Li content in the octahedral and tetragonal sites of the crystal.

Figure 16 shows the ^7^Li and ^6^Li spectra of tetrahedral LLZO (t-LLZO) before and after ^6^Li enrichment. It is immediately apparent from the MAS spectra that there was a significant increase in the resolution of ^6^Li spectra when prepared with enriched ^6^Li, as is evident by the now-detected presence of a small peak near 0.1 ppm attributed to Li_2_CO_3_ from the synthesis process.

Figure 17 shows the ^6^Li spectra of LLZO after being exposed to air, which allowed protons to enter the sample. After protonation, the authors determined from the ^6^Li spectra of Li_3.8_H_3.2_La_3_Zr_2_O_12_ the presence of three distinct Li peaks belonging to three different sites in the crystal located at 1.6, 1.2, and 0.6 ppm, which they extrapolated back to sites in the original LLZO lattice. By comparing the intensities of peaks before and after protonation, the authors were able to deduce that the 1.2 ppm band belonged to Li ions in distorted octahedral sites, on account of previous models suggesting lower activation energies required for Li-proton exchange [61]. Though not as straightforward, the authors also attributed the 0.6 ppm band to the tetragonal site and the 1.6 ppm band to another, less distorted octahedral site [52].

To study the dynamics of Li ions in these sites, the group also performed T_1_ inversion recovery experiments, and found that the octahedral sites exhibited a greater mobility than the tetragonal site, which was expected, further strengthening their assignments. Finally, ^6^Li-^1^H CP MAS spectra revealed motional narrowing occurring around 70 °C and deduced that the Li ions exchange between octahedral and tetragonal sites.

NMR relaxation studies are also extremely useful for studying Li ion dynamics and probing self-diffusion, especially when PFG measurements cannot be performed, as is often the case in solid electrolytes. Bottke et al. previously studied ^7^Li relaxation in Al-doped LLZO and observed anomalous relaxation behavior; depending on whether the measurements were performed in the lab frame or in the spin-locked rotating frame, they noticed vastly different activation energies ranging from 0.1 eV up to 0.4 eV [62]. Wondering if this was unique to this system, the authors performed relaxation measurements again, this time on Li_6.5_La_3_Zr_1.75_Mo_0.25_O_12_ (LLZMO) crystals [63].

Figure 18 highlights the lineshapes of the ^7^Li spectra as a function of temperature. From the lineshapes, the authors discovered a near homogeneous motional narrowing, compared to Al-doped LLZO. From the intensities of the quadrupolar interactions (~300 kHz) in the lineshapes, and the subsequent vanishing at higher temperatures, the authors also deduced that the Li ions were extremely mobile, with mean jump rates of at least 10^6^ s^−1^. By performing SLR measurements in the lab and rotating frame of reference, they found the same mean activation energy of ~0.29 eV, lower than that of tetragonal LLZO [62,64], suggesting high Li ion diffusion. From these plots, they were also able to estimate the Li self-diffusion coefficient using the Einstein–Smoluchowski (ES) relation [65] to be about 10^−15^ m^2^/s.

Additionally, the authors employed spin alignment echo (SAE) spectroscopy to further validate the measured activation energy [64,66], the results of which are plotted in Figure 19. By using a preparation time of 10 μs and arraying the mixing time between 10 μs and 10 s, the authors were able to probe different Li ion dynamics by observing the corresponding echo amplitude, including slow jump processes between sites with different electric field gradients (EFGs), spin diffusion, or ordinary spin lattice relaxation. The authors expected that the jump process operated on a slower time scale than diffusion or relaxation, leading to a two-step amplitude decay, and this was indeed seen in experiment. The activation energies deduced from these amplitudes were also in agreement with the previously calculated activation energy of 0.3 eV.

In particular, at low temperatures and high preparation times, the spin echo amplitude decayed to a plateau, which the authors attributed to a Li ion jump process between 96 h sites of the LLZMO lattice, through a secondary jump to the 24 d site [67,68].

Other groups have attempted to rectify discrepant measurements of Li ion dynamics in the garnets. As has been explained in this work, T_1_ relaxation, lineshape analysis, PFG NMR, and other experiments can be used to independently deduce Li ion self-diffusion coefficients [65,69,70,71,72,73]. Many of these experiments were performed on polycrystalline samples, due to the relative ease of synthesizing the large samples necessary for NMR. However, the measured diffusion coefficients have varied greatly, in a manner not seen in liquid or homogeneous electrolytes, ranging from 10^−13^ m^2^/s to 10^−18^ m^2^/s at room temperature. To try and rectify this discrepancy, Dorai et al. employed PFG NMR experiments on single crystalline and powder Li_6.5_La_3_Zr_1.5_Ta_0.5_O_12_ (LLZTO) [74]. The standard stimulated echo pulse sequence, as shown in Figure 20, was used, comprising three $\frac{\pi }{2}$ pulses and two intermediate gradient pulses. In an attempt to minimize any magnetic field inhomogeneity effects, which could also interfere with diffusion coefficient calculation, the aforementioned gradient pulses were not pure square pulses but rather a composite of a Quarter-sine pulse, followed by a standard square pulse, ending with a linear ramp down to minimize eddy current effects from fast ramping.

The measured single crystal diffusion coefficients from NMR were on the order of 10^−13^ m^2^/s, which agrees with earlier works on similar LLZTO materials. A clear discrepancy appeared upon comparison of NMR diffusion and EIS-calculated diffusion. Computing the Haven ratio, which is simply the ratio of NMR diffusion to EIS diffusion, to be 0.4, this discrepancy is thought to be due to correlated Li ion motion in the material, rather than pure Brownian motion [75]. Molecular dynamics simulations on this material have yielded the same ratio, and suggested that this correlated motion comes from the combined migration of Li ions in tetrahedral and octahedral sites [76]. In addition, the authors attempted to see whether changing the time scale of the diffusion experiment changed the value of the diffusion coefficient, which would imply differing Li ion dynamics on different time scales. However, although the diffusion time was arrayed from 20–600 ms, as shown in Figure 21, there was no significant change in the measured diffusion coefficient.

Comparing the diffusion in the single crystal to that in the powder, it was clear that the powder exhibited slower overall Li ion diffusion. In addition, there was a more significant time dependence seen in the powders that was not seen in the single crystals, which could be due to the grain boundaries induced by the powder, which restricted Li ion mobility.

## 7. NASICON Electrolytes

Sodium superionic conductor (NASICON)-type electrolytes are another attractive family of crystalline electrolytes, with the general structure AM_2_X_3_O_12_, where A is an alkali metal, M is either Ti or Ge, and X is either Si or P. Classic NASICON materials, such as LiTi_2_(PO4)_3_ (LTP) and LiGe_2_(PO_4_)_3_ (LGP) were potential electrolyte candidates, as LTP can exhibit room temperature conductivities up to 10^−5^ S/cm, and LGP is known to be stable against Li up to 6 V [77]. Attempts to improve upon this conductivity have shown that partial substitution of Li^+^ with Al^3+^, forming Li_1.5_Al_0.5_Ge_1.5_(PO_4_)_3_ (LAGP), yields such a result [78,79].

Vyalikh et al. discovered a five-fold increase of the ionic conductivity of LAGP through the addition of yttrium oxide, forming LAGPY [80]. X-ray diffraction and electron microscopy studies on LAGPY suggested that the YPO_4_ lies at the LAGP grain boundaries, allowing for faster ion mobility through said boundaries. Employing ^7^Li lineshape analysis, relaxation experiments, and diffusometry studies, they attempted to understand the nature of Li-ion dynamics and how it is affected by the crystal geometry. From lineshape analysis performed on LAGP and LAGPY, the authors calculated correlation times and Li ion jump rates, which were both on the order of 10^−4^ s^−1^, as shown in Figure 22. This suggests that Li ions were not hopping from site to site faster from the addition of Y_2_O_3_. Repeating this measurement using field cycling NMR, the authors found no significant change in the jump rates and calculated a mean of 0.37 eV for both materials.

It was also possible to probe ionic motion through the use of PFG NMR. VT diffusion measurements on LAGPY revealed an activation energy of 0.32 eV, which appears to agree with FC Measurements. Through the Einstein–Smoluchowski relation [65], taking an average jump length to be the distance between Li1 and Li3 sites and a room temperature D ~ 10^−13^ m^2^/s, the authors calculated a jump rate an order of magnitude higher than found from relaxation measurements, which suggested that LAGPY exhibited significant dynamical heterogeneity.

Finally, through DFT calculations, the authors deduced that the addition of Y_2_O_3_ changed the concentration of Li ions occupying Li1 and Li3 sites; the increased concentration of Li ions in the Li3 sites had the effect of pushing ions in Li1 sites to other Li3 sites, lowering the hopping energy barrier. Thus, NMR and DFT showed that the introduction of Y_2_O_3_ into LAGP led to a denser intergrain microstructure [81], while electron microscopy showed improved grain boundaries, leading to the aforementioned increase in conductivity.

## 8. Conclusions

This review has shown only a few of the ways, using selected representative examples, in which nuclear magnetic resonance can be utilized to probe solid inorganic electrolyte systems and extract crucial structural and dynamical information. The wide applicability of NMR to most materials, and the time/length scales which it can probe, further demonstrate the power of the technique. NMR spectroscopy as an analytical tool will surely prove invaluable in aiding the search to produce a viable all solid-state battery over the coming years.

## Figures and Tables

**Figure 1 ijms-21-03402-f001:**
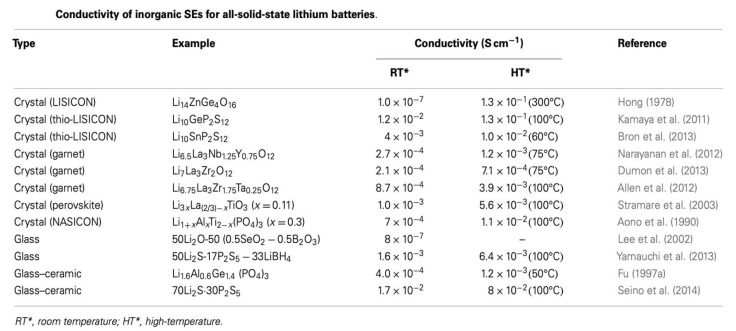
Various types of solid electrolytes and reported conductivities. Courtesy of Cao et al. [2].

**Figure 2 ijms-21-03402-f002:**
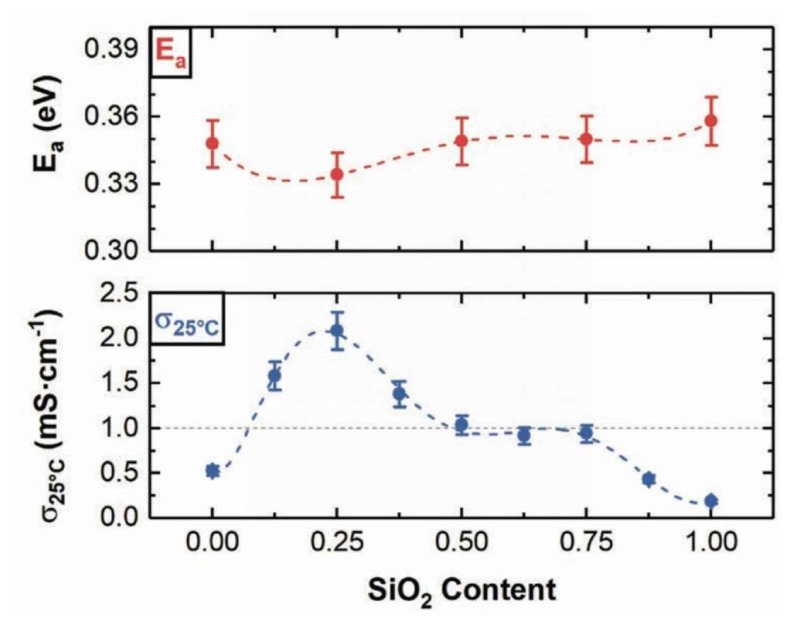
(Top) Activation energy and (bottom) room temperature ionic conductivity of aLi_2_S-γB_2_S_3_-xSiO_2_-zLiI (LIBOSS) with varying silica content [11].

**Figure 3 ijms-21-03402-f003:**
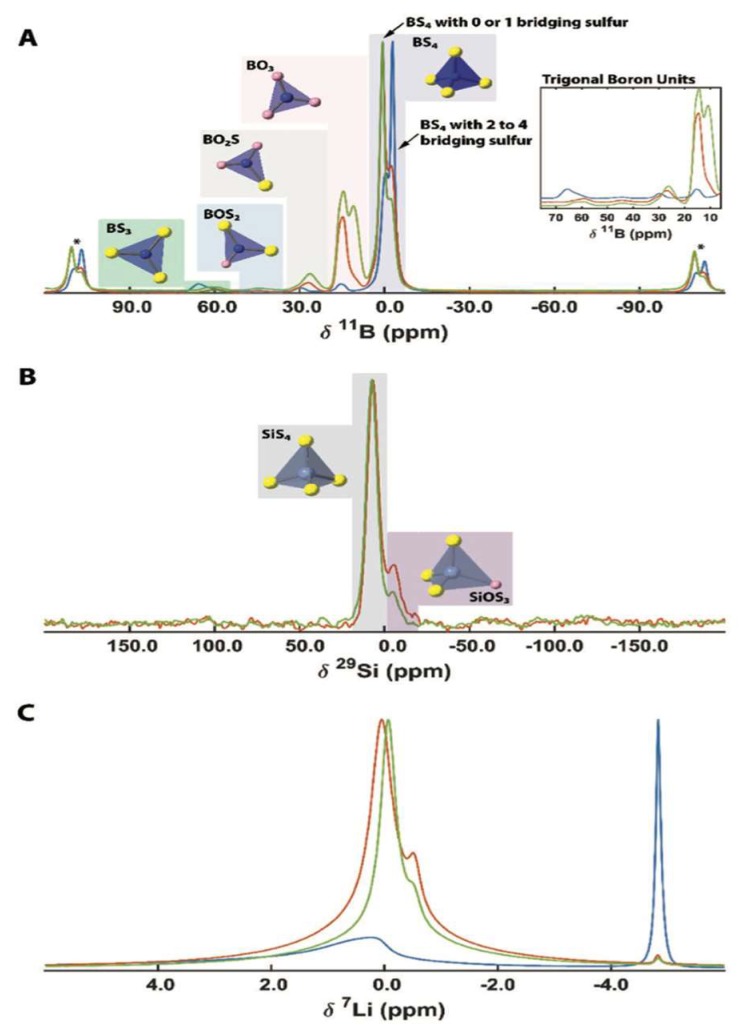
(**A**) ^11^B, (**B**) ^29^Si, and (**C**) ^7^Li magic angle spinning (MAS) spectra of LIBOSS for x = 0 (blue), x = 0.25 (orange), and x = 0.5 (green) [11].

**Figure 4 ijms-21-03402-f004:**
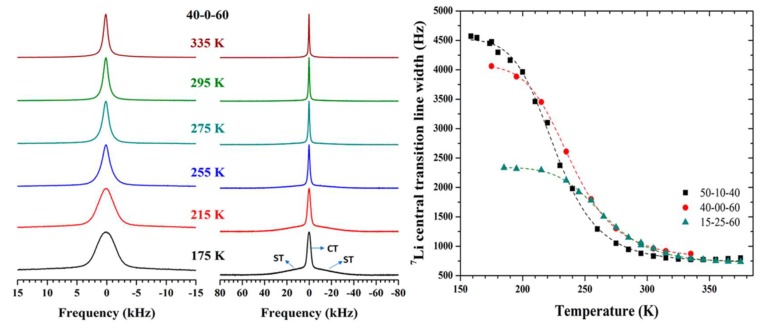
(Left) Static ^7^Li NMR spectra with temperature in the 40–60 Li_2_S–GeSe_2_ glass. (Right) Motional narrowing of both the central transition (CT) peak is clearly seen at elevated temperatures [16].

**Figure 5 ijms-21-03402-f005:**
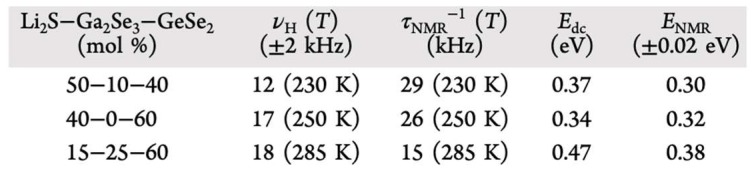
EIS and NMR-derived ion jump frequencies and activation energies for three LGGS glasses [16].

**Figure 6 ijms-21-03402-f006:**
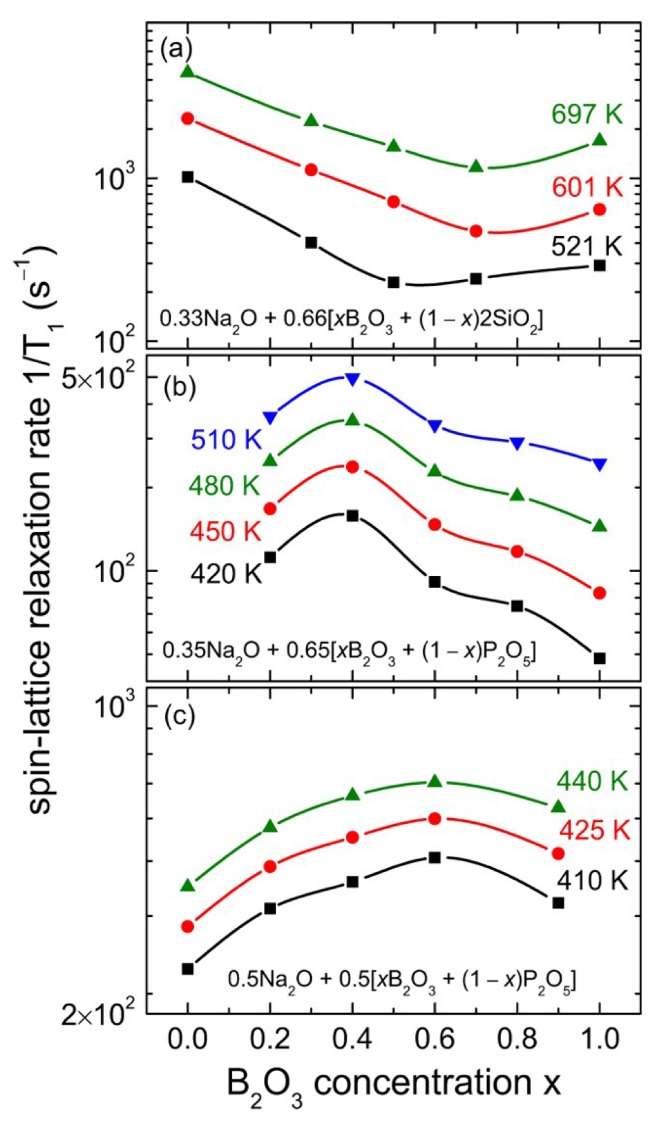
^23^Na Relaxation data as a function of Boron concentration in (**a**) borosilicate glasses and (**b**,**c**) various borophosphate glasses [21].

**Figure 7 ijms-21-03402-f007:**
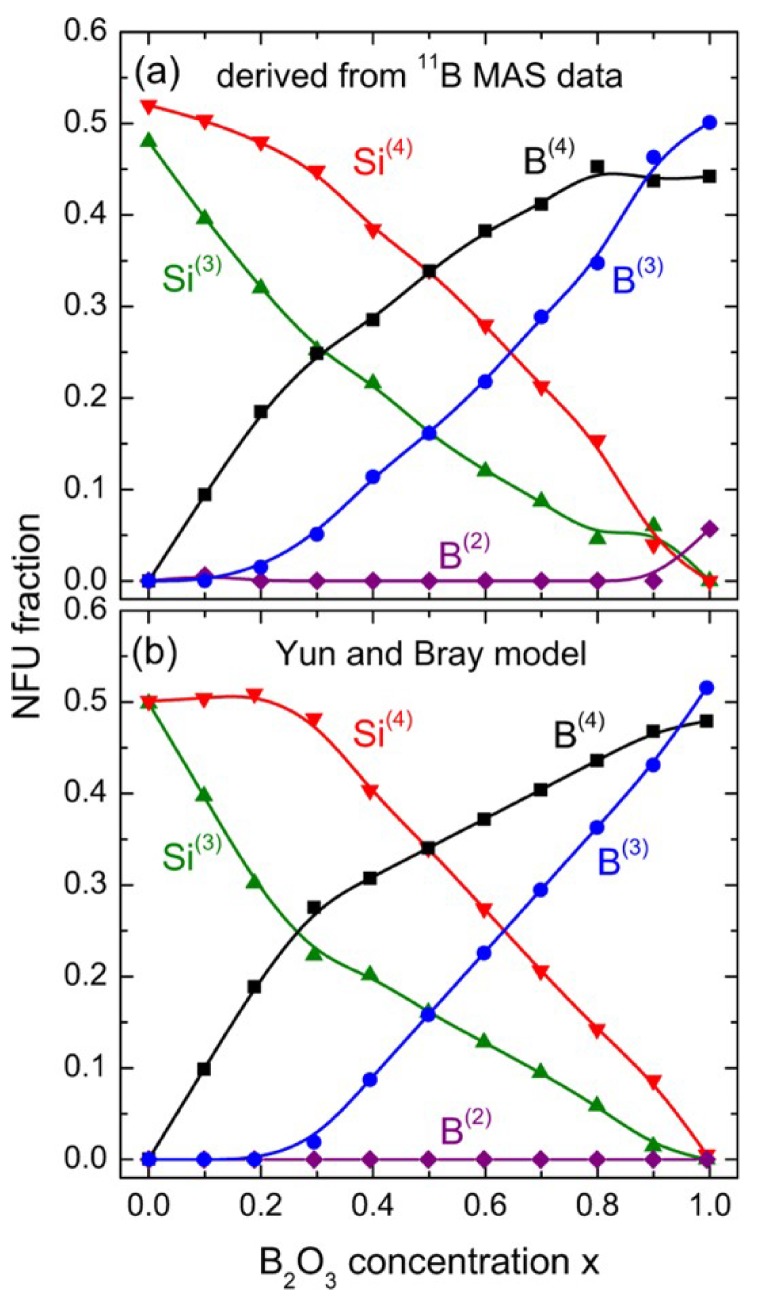
Network forming unit (NFU) fractions of sodium borosilicate glasses as determined (**a**) by ^11^B NMR, and (**b**) by the Yun and Bray model [21].

**Figure 8 ijms-21-03402-f008:**
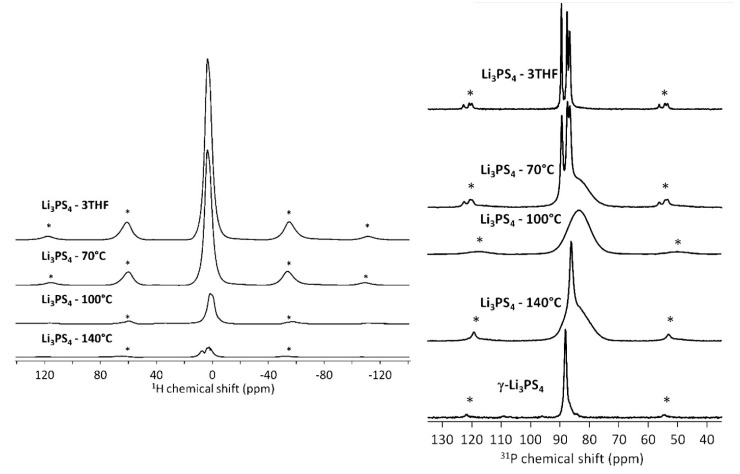
(Left) Proton MAS spectra of the Li_3_PS_4_ glasses treated at various temperatures. (Right) ^31^P MAS spectra of the same Li_3_PS_4_ glasses. The asterisks indicate spinning sidebands [30].

**Figure 9 ijms-21-03402-f009:**
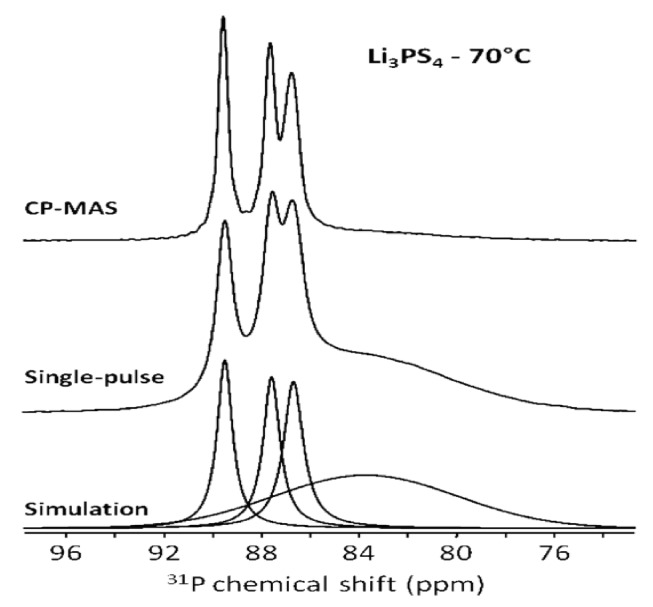
^31^P cross-polarization (CP) and single-pulse spectra for the Li_3_PS_4_ glass treated at 70 °C. The broad peak around 84 ppm does not interact with the THF and thus does not appear in the CP spectrum [30].

**Figure 10 ijms-21-03402-f010:**
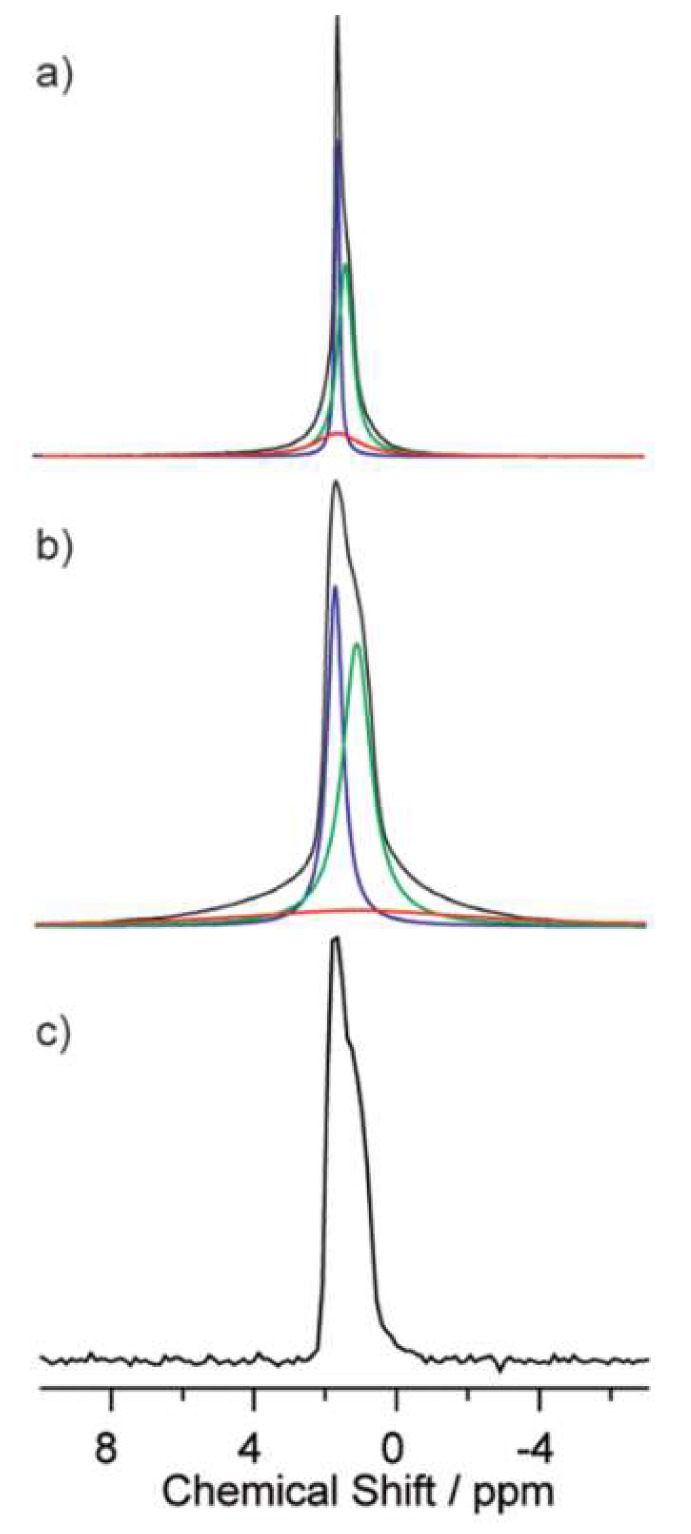
^7^Li MAS NMR spectra of 70 gc at (**a**) 360 K and(**b**) 270 K. (**c**) ^6^Li MAS NMR spectrum of 70 gc at room temperature [36].

**Figure 11 ijms-21-03402-f011:**
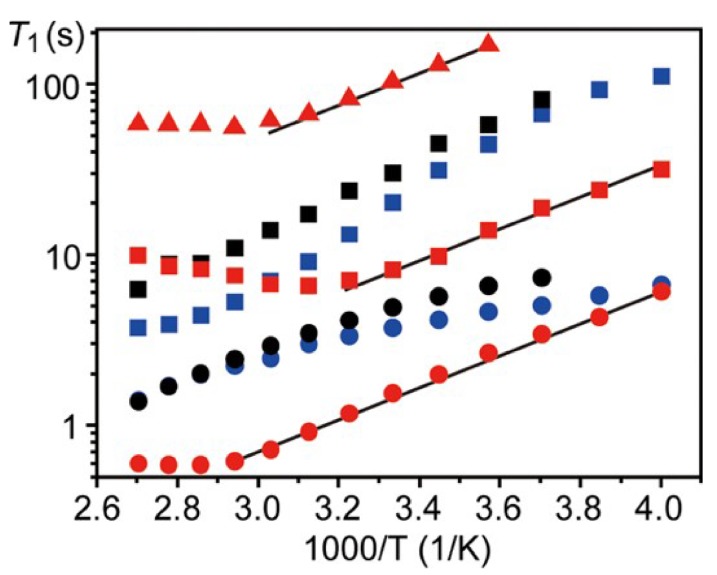
Plots of ^6^Li (triangle), ^7^Li (circle), and ^31^P (square) T_1_’s for 70 gc (red), 75 gc (blue), and 70 g (black). The solid line through ^7^Li T_1_ data points for 70 gc is the least-squares fit one to a straight line [36].

**Figure 12 ijms-21-03402-f012:**
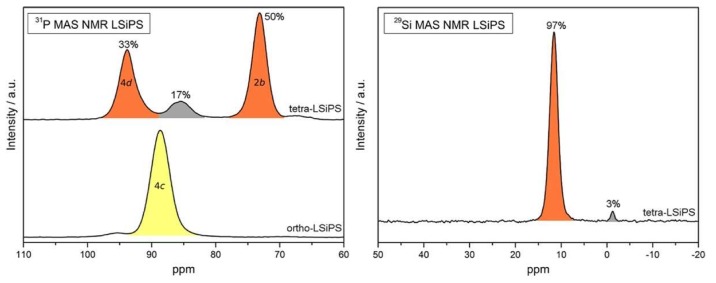
(Left) ^31^P NMR spectra of tetrahedral (top) and orthorhombic (bottom) LSiPS. (Right) ^29^Si NMR spectra of tetrahedral LSiPS [41].

**Figure 13 ijms-21-03402-f013:**
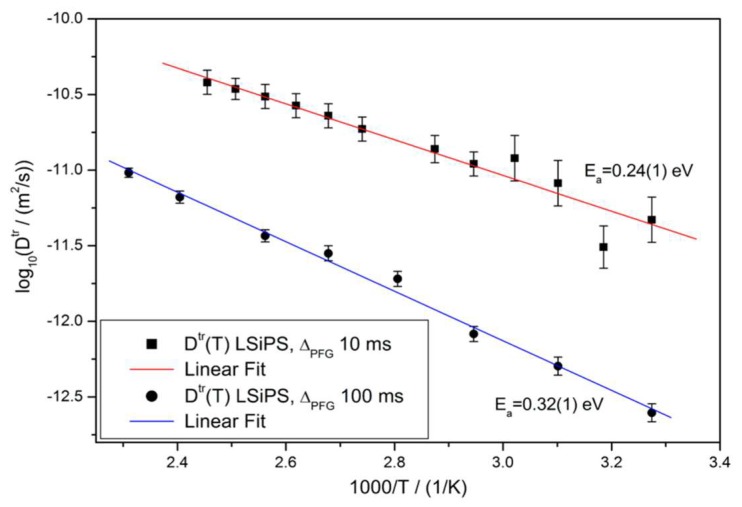
Arrhenius plots of variable temperature diffusion in tetra-LSiPS, at both 10 ms and 100 ms diffusion time [41].

**Figure 14 ijms-21-03402-f014:**
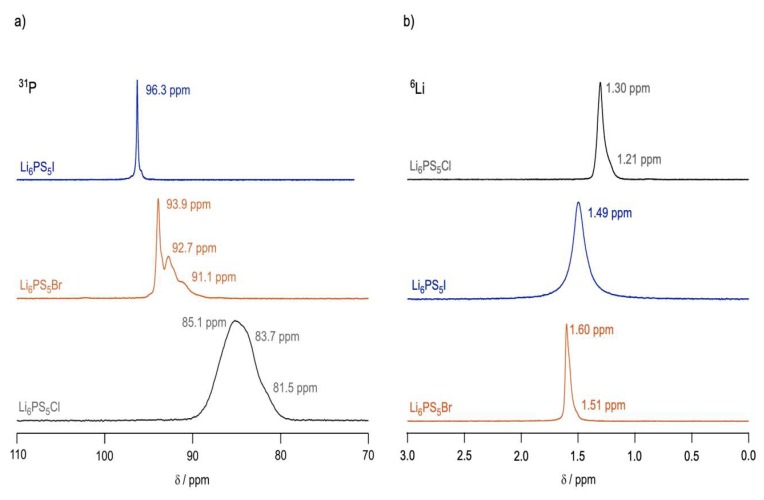
(**a**) ^31^P and (**b**) ^6^Li MAS spectra for all three argyrodites [46].

**Figure 15 ijms-21-03402-f015:**
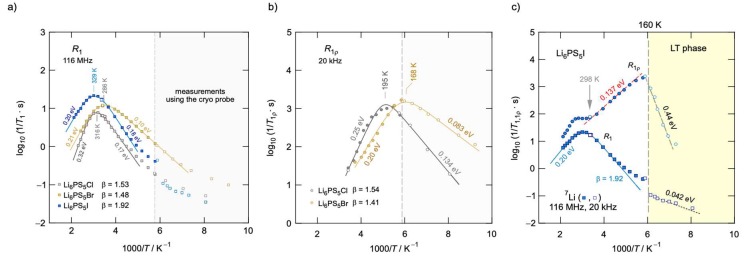
Arrhenius relaxation plots of Li_6_PS_5_X in the (**a**) lab frame of reference, (**b**) rotating frame of reference of 20 kHz, and (**c**) combined relaxation plots of Li_6_PS_5_I [46].

**Figure 16 ijms-21-03402-f016:**
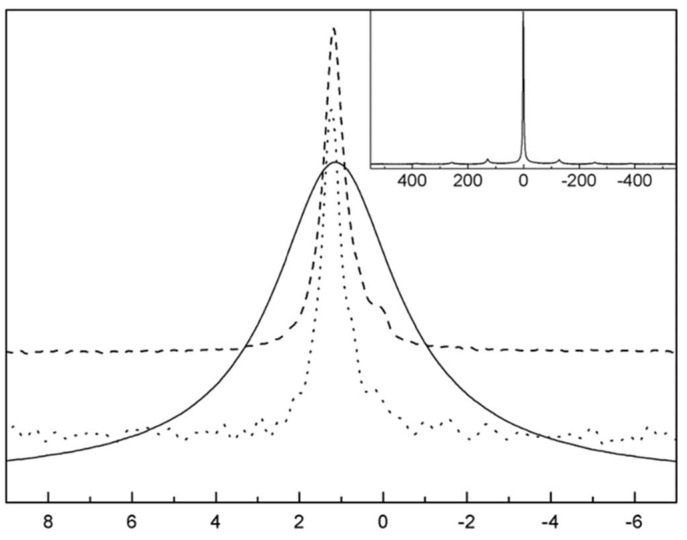
^7^Li (solid) and ^6^Li (dotted) MAS spectra of t-LLZO prepared using natural abundance Li_2_CO_3_. The same ^6^Li spectra (dashed) is also presented when prepared with enriched ^6^Li [58].

**Figure 17 ijms-21-03402-f017:**
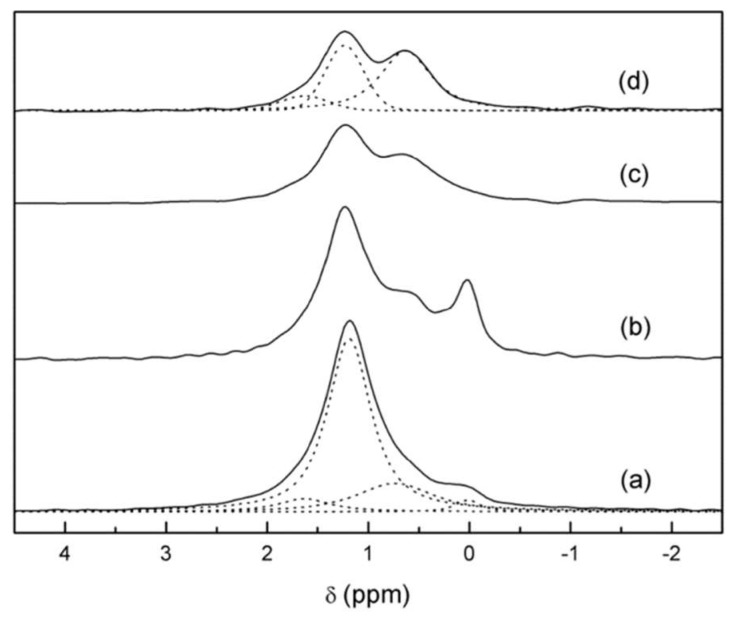
^6^Li MAS spectra of (**a**) enriched LLZO prepared as is (**b**) protonated LLZO before and (**c**) after washing of impurities, and (**d**) with proton decoupling. The dotted lines represent the deconvolution fit [58].

**Figure 18 ijms-21-03402-f018:**
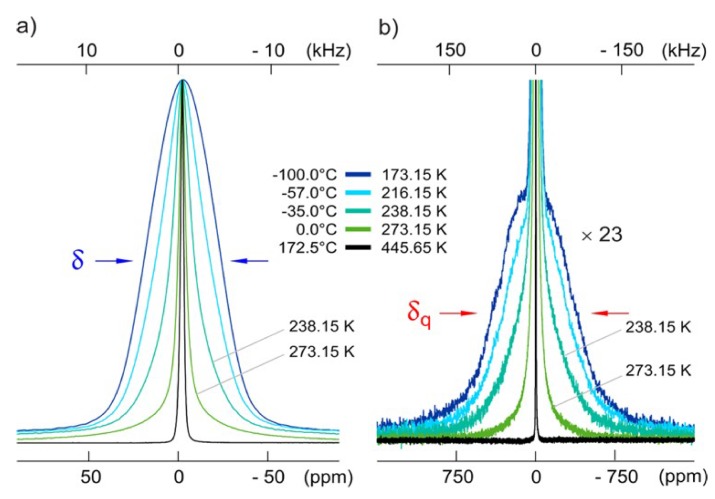
(**a**) Single Pulse ^7^Li spectra of Li_6.5_La_3_Zr_1.75_Mo_0.25_O_12_ (LLZMO) with temperature. (**b**) The same temperature dependent spectra zoomed in to see the averaging of quadrupolar effects [63].

**Figure 19 ijms-21-03402-f019:**
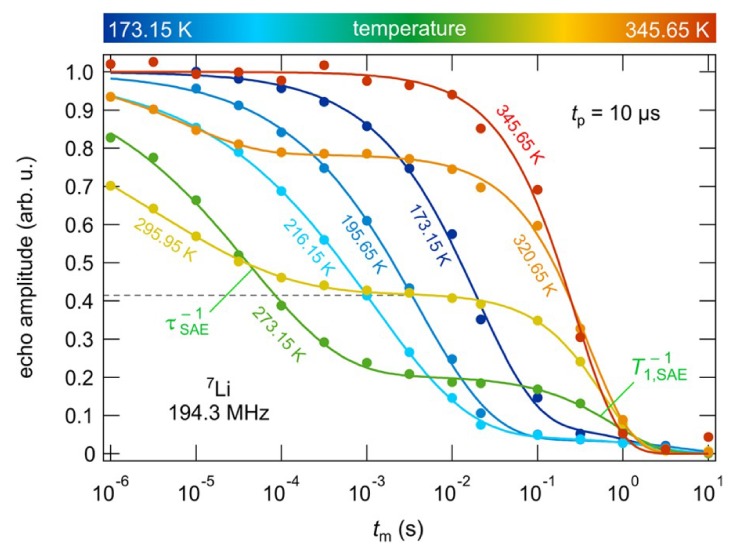
Two-time correlation functions measured from SAE experiments, showing two-step decay [63].

**Figure 20 ijms-21-03402-f020:**
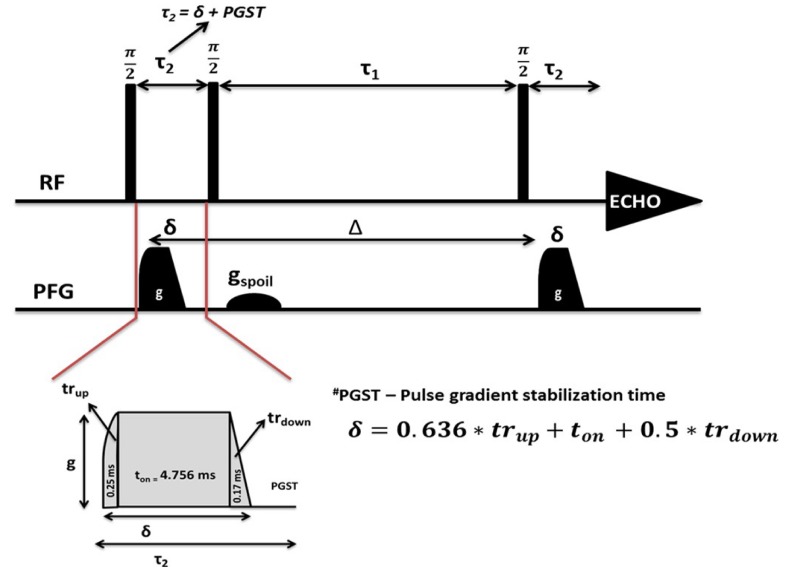
PFG stimulated echo pulse sequence, with composite gradient pulses [74].

**Figure 21 ijms-21-03402-f021:**
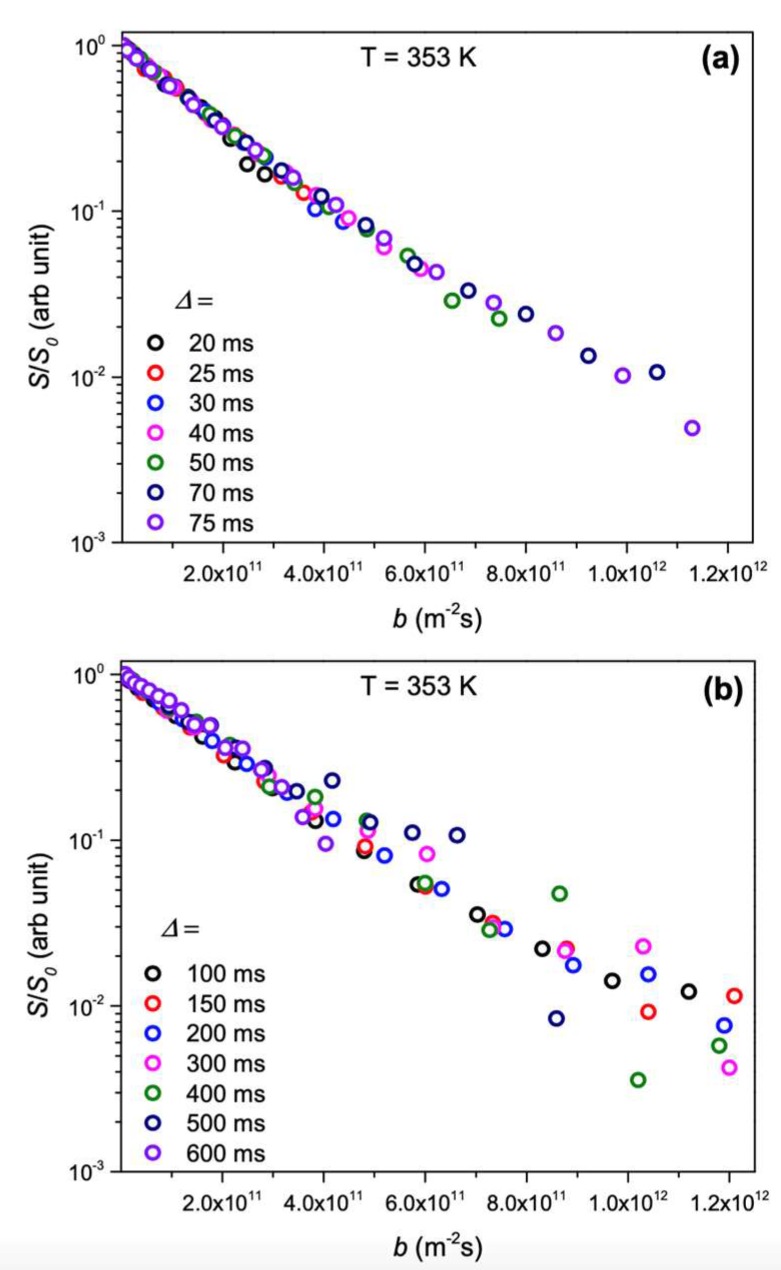
Signal Attenuation plots for single crystalline Li_6.5_La_3_Zr_1.5_Ta_0.5_O_12_ (LLZTO) at diffusion times (**a**) from 20–70 ms and (**b**) from 100–600 ms [74].

**Figure 22 ijms-21-03402-f022:**
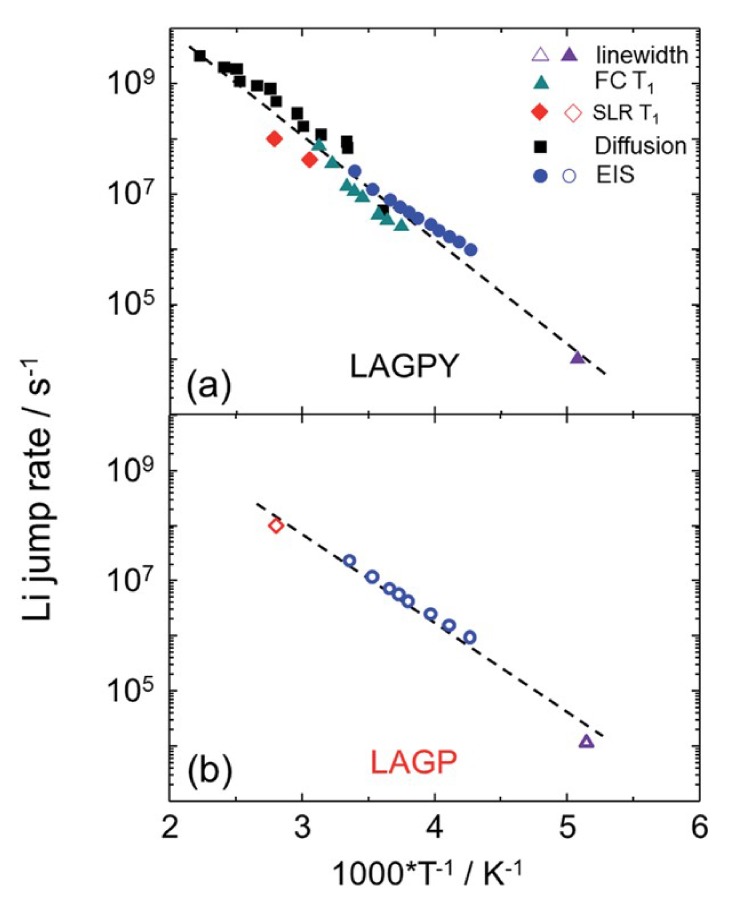
Plots of Li jump rates with inverse temperature for (**a**) LAGPY and (**b**) LAGP, recorded using a combination of linewidth analysis, spin lattice relaxation (SLR) measurements, FC NMR, PFG NMR, and EIS [80].

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
