# Peer review of "NMR Investigations of Crystalline and Glassy Solid Electrolytes for Lithium Batteries: A Brief Review"

_ijms, 2020, doi:10.3390/ijms21093402_

Round 1

Reviewer 1 Report

The review begins with a basic summary of the equations used in various measurements. This section could be improved by the addition of some references to standard works on these methods. The authors should probably use the more general Solomon equation for 1/T1. References to the solid-state NMR techniques employed in the various papers described would also be very helpful to readers unfamiliar with solid-state NMR.

The main part of the review covers a number of studies involving glassy, glass-ceramic, argyrodite, garnet and NASICON electrolytes. Each type is describe using a good number of examples with a number of figures extracted from the chosen papers.

There seems to be some confusion at the beginning of section 3 over whether glassy or glass-ceramic are being discussed. A clear distinction between these two classes at this point would be helpful.

The review is generally quite well written but there are certain passages which could be better phrased, such as:

“because the lack of grain boundaries, making it an attractive electrolyte candidate” (line 88)

 “with the isolated boron tetrahedra being more present” (line 114)

“In addition, this amorphous only appears at samples prepared at intermediate temperatures” (line 343)

Errors include:

has → have (line 49)

was → were (line 110)

affects → effects (line 152)

grain boundaries → grain-boundary (line 225)

° → °C (line 252)

Argyrodites → Argyrodite (line 370)

Formatting in references 62 and 65.

Formatting of several chemical formulae is required (lines 236 & 309)

Figure 7 is low quality and should be improved if possible.

Figure 9 is distorted.

Author Response

Thank you for your comments and suggestions.

  • We have replaced the equation for 1/T1 with the Solomon equation, and introduced the spectral density function. We have also added references to NMR texts and papers, which elaborate on the techniques discussed in the introduction. The references added in the revised manuscript are #'s 3-6.
  • In section 3, we reworded the initial paragraph to clearly denote glass-ceramics (GC’s) as separate from pure glasses, and made clear that GC’s were the main subject of this section. To further this point, we changed “simultaneously exhibiting both glassy and crystalline phases” to “simultaneously exhibiting both amorphous and crystalline phases” (Lines 230-231).
  • Concerning the formatting of, and typos in the text, we have made all the recommended adjustments.
  • We have rephrased each highlighted statement to make them flow more smoothly.
  • The resolution of Figure 7 has been improved slightly.
  • Figure 9 has been resized, and will no longer appear distorted.

Reviewer 2 Report

The manuscript by Morales and Greenbaum describes the possibilities offered by NMR to the structural and transport properties of lithium-based electrolytes. 

The review is comprehensive, but I would suggest to touch the following points: 1) FC relaxometry is mentioned, but no details or references are given; 2) structural elucidation through paramagnetic effects should be mentioned (see the work by Clare Grey and Guido Pintacuda) 3) mobility measurements performed with tailored equipment have been pioneered by Clare Grey and Elodie Salager, and they are definitely worth mentioning.

Author Response

Thank you for your comments and suggestions.

  • We have added references to Field Cycling NMR (5,6 in the revised manuscript), and elaborated on the use of the technique in finding the frequency dependence on the relaxation time T1.
  • We have reviewed Clare Grey’s works on NMR of paramagnetic battery materials, but since much of this work pertains to electrode materials, we felt that her works were outside the scope of our discussion on solid electrolytes.